# Church and State and the Marital Rights of Old Believers in Latvia: From Illegality to Secularization

**Maija Grizāne** 

Institute of Humanities and Social Sciences, Daugavpils University, LV-5401 Daugavpils, Latvia; maija.grizane@gmail.com

**Abstract:** The paper analyses state religious policy in different historical periods and its impact on the development of religious doctrines about marriage within the Old Believer community of Latvia. Based on published and unpublished historical sources, legislative acts, periodicals, and data from existing Old Believer parishes, it is clear that state policy concerning religious minorities greatly influenced the development of the Old Believer community. Old Believer marriages were not recognized by the Russian Orthodox Church until 1874 when the first possibility of obtaining legal marital status was introduced as registering families in police register books. The Old Believers of Latvia, who originally belonged to the Fedoseevcy denomination and denied any family life as such, registered their families quite rarely. However, during the first decades of the 20th century, the Fedoseevcy of Latvia adopted the teaching and marriage ceremonies of the Pomorians. By the interwar period, Old Believer marriages were legalized by both the community and the state. Soviet secularization further facilitated the development of secular marriage unions with followers of other confessions. In the present day, Old Believer religious marriages are legalized in Latvia, though the number of them has decreased as Old Believers are most likely to choose civil marriage registrations.

**Keywords:** Old Believers; Latvia; church and state relations; marriage; family; religious minority

## 1. Introduction

The Old Believers are one of the most conservative groups in Latvian society, as their identity and cultural traditions are completely reliant on the past. This feature has been prevalent since the emergence of the Old Believers movement, when, in their view, the Orthodox faith was divided into pre- and post-reform communities. In the first half of the 17th century, the patriarch of the Russian Orthodox Church, Nikon, initiated a return to the origins of Christianity and sought to eliminate the accumulated errors in the interpretation of teaching and practice. With the support of the tsar and the patriarchs, in 1667, it was determined that the liturgical books needed to be corrected, and this became a fateful decision in the splitting of the Russian Orthodox Church. Most of the Orthodox priests joined the reformed Church, and those who did not accept the innovations—the schismatics (*raskolniki*), later called the Old Believers—were left without priests[1] and, consequently, without traditional religious practice with sacraments and church services. Thus, the Old Believers severed ties with the Orthodox Church and, despite statements about preserving the old order, had to adapt to new circumstances, organize new forms of religious life, and creatively fill in the missing elements of faith. As the Russian emigrant historian Sergej Zen'kovskij (1907–1990) explained, Old Believers claiming preservation of the old faith had to create new "old" faith and "old" rituals that were not based on the Orthodox Church hierarchy (Zen'kovskij 2006, p. 254). Accordingly, the Old Believers, who call themselves the Old Orthodox, turned to the past to seek role models and the ideals of the true Christian faith. Latvian researcher of the Old Belief Nadezda Pazuhina has pointed out that the relationship of Old Believers with the past is determined not by official historical discourse or collective memory but by belonging to a "living tradition" and its real experience

(Pazuhina 2006); the followers of the Old Belief were forced to adapt to existence outside the church system and became accustomed to relying on past experience while inevitably inventing new forms of religious practice.

The state, which declared Russian Orthodoxy a national religion and an important domestic political tool for the unification of the multicultural empire, perceived the Old Believers as enemies of autocracy and heretics who had to be "returned" to the Russian Orthodox Church. In turn, the Old Believers declared the state authority as obsessed with the "spirit of Antichrist" and began looking for new territories to avoid persecution and punishments (Farmakovskij 1866). The newly created local communities were managed on the model of peasant meetings (*shodi*), during which decisions were made (Shhapov 1862, p. 75). Over time, variants of doctrines, commonly called denominations or concords (*soglasie*), have formed. Their names were based on the region of origin (e.g., the Pomorians) or by the name of the founder (e.g., the Fedoseevcy). Each community of Old Believers established its own domestic order that contributed to the economic consolidation and independence of the communities from centralized management policies. The most senior representatives of the communities—the elders—became the guardians of the creed. At the beginning of the Old Believer movement, the elders formulated the main ideas of the Old Believer worldview and its theological positions. The immutability of rituals[2] and theologically determined inability to change anything in the liturgical books or the order of worship took a central position. The rituals were declared as dogma, contradicting the position of the Russian Orthodox Church, which argued that rituals had to serve the needs of believers, not the other way around. This confrontation between the Russian Empire (state) and the Russian Orthodoxy (church), on the one hand, and the followers of the Old Belief, on the other, led to a split within the Russian people and the long-lasting isolation of a significant portion of the population. Several internal movements within the Old Belief, designated as sects by the state authorities, competed with each other to prove their authenticity and the right to represent the pre-reformed Orthodoxy. The most influential Old Believer denominations on the territory of the Baltic region were priestless Bespopovcy[3]: the Pomorian and the Fedoseevcy. The Pomorian originated from the monastery on the Vyg River in the Pomor'e region (present-day Karelia) and was supported by Moscow parishes, while the Fedoseevcy was founded in 1696 by Feodosij Vasil'ev (1661–1711) in Nevel, which is present-day Lithuania. Originally, the Old Believers of Latvia, Lithuania, and Estonia practiced the Fedoseevcy teaching; in the current day, with some exceptions, all parishes are Pomorian. Along with the state's religious policy and legislation regarding the Old Belief, this transition was prompted by the issue of marital rights.

The paper analyses church and state relations and the influence of religious policy on the views of marriage within the Old Believer community of Latvia from the second part of the 19th century to the present day. By comparing different historical periods, it is possible to follow the adaptations of the Old Believers regarding the changing political and social conditions and their degree of influence on an initially closed and very conservative religious group.

## 2. Studies on the Old Believers and Religious Freedom: An Overview

The Old Believers movement was legitimized during the second part of the 19th century and the first decade of the 20th century. During this time, new laws were passed that gradually made social and religious life easier for Old Believers. In the Russian historiography of this period, there are two opposing opinions: the representatives of the Russian Orthodox Church insisted on the Old Believers' threat to the existing order, while some representatives of the intelligentsia admired the Old Believers as the owners of the "true spirit of the Russian people". Afanasij Shhapov (1831–1876), a Russian historian and ethnographer, attributed Church–civil democracy to the Old Believers and considered it "a bold protest" against the reformation of the state following the example of the Western model involved by Peter I (Shhapov 1859). In turn, Nikolai Leskov (1831–1895), a Russian

writer, who visited the Old Believer parish of Riga in 1862, paid attention to the Old Believer confrontation with the state and the ideological division within the Old Belief that led to an intense search for identity (Trofimov 2006). Official studies on the Old Believers were obviously influenced by the state religious policy and were dedicated to the history of certain Old Believer groups (Prugavin 1881; Gromoglasov 1895; Pravda o russkih raskol'nikah 1897; Zhuravljov 1794; Kratkaja istorija raskola 1866; Nikiforovskij 1892; Subbotin 1892). Another group of publications, which were highly polemical, provided practical information to Orthodox missionaries to use against Old Believers (Ivanovskij 1883; Iz praktiki svjashhennika-missionera 1900; Kak prezhde obrashhali raskol'nikov 1882; *Missionerskoe obozrenie* 1896–1908). Finally, a third group of publications consisted of propagandistic examples of Old Believers' conversion to Orthodoxy and consequent satisfaction (Novgorodskij 1888; Rasskazy byvshih staroobrjadcev 1892). In general, the publications contain one-sided interpretations of the Old Belief as the heretical movement of sectarians and the Russian Orthodox Church as the undeniable authority for every Russian person.

Western authors were interested in the Old Belief as a unique phenomenon of the Russian people; it was often described from the historical perspective of church–state relations based on the authors' own observations and the translations of Russian authors (Franck 1859; Heard 1887; Miller 1907). The Western authors emphasized the social and political consequences of the Old Belief and its complicated relationship with state institutions. Especially notable is the study conducted by the American diplomat Albert Farley Heard (1833–1890), who served as a representative of the Chinese government in Russia and published "The Russian Church and Russian Dissent, Comprising Orthodoxy, Dissent, and Erratic Sects" in 1887. This work was reprinted several times, including as recently as 2017, and still retains its relevance in Western historiography as an example of the history of Russian Orthodoxy. Based on the publications of Western authors (in French, English, and German) and translations of Russian authors (for example: Mouravieff 1842; Palmer 1871; Stanley 1861; Gagarin 1872), Albert Heard described in detail the development of the Russian Orthodox Church as a powerful authority, as well as the influence of certain persons and events on this process. He named the schism of the Russian Orthodox Church "the Raskol" by transliterating the Russian term, an approach that is followed to this day.

We are not aware of any studies conducted on Latvian Old Believers in the 19th century, with the exception of ethnographic and statistical surveys (Sementovskij 1872; Obzor Vitebskoj gubernii 1894; *Pamjatnaja knizhka Vitebskoj gubernii* 1891–1906). In the 19th century, the eastern territory of Latvia belonged to the Vitebsk province of the Russian Empire, supported by the Russian Orthodox Church, while the western part belonged to the Baltic province with the predominant Protestant German opinion found in the periodical publications of "Rigasche Zeitung". In the latter's view, Old Believers, called schismatics (Schismatiker), had to be returned to the path of sense and enlightenment (auf den Weg der Vernunft und Aufklärung zurückführen)—that is, to the Russian Orthodox Church (Moskau 1869; Die Wirksamkeit des heiligen Synods 1876). The position of the local Latvian press was equally radical: Old Believers were described as heretical fanatics who were dangerous to local people and the Russian Orthodoxy (Iekšzemes atskats 1886; Iekšzemes ziņas 1886; Sv. sinoda virsprokurora 1886). This attitude can be explained by the dependence of local periodicals on state authorities.

The first Latvian Old Believer scholar was Ivan Zavoloko (1897–1984), a leader of the Latvian Old Believer community who also had great authority in neighboring Lithuania and Estonia. To this day, Zavoloko is an authority for many Old Believers of the Baltic states. He collected Old Believer books and folklore, published articles on doctrinal issues, and developed the first schoolbook on the Old Belief (Zavoloko 1929, 1933, 1935, [1933] 1936, [1933] 1937, 1937, 1939). His works on church history argued that the history of the Old Belief should not be seen as beginning in the mid-17th century but from the very emergence of Christianity. In his interpretation, the Orthodox Church and the Russian state acted as traitors to the Russian faith, and, accordingly, all state persecution of Old Believers is

special "suffering for the faith" (Zavoloko 1937). Zavoloko also developed the idea that the Old Belief appeared on the territory of Latvia long before the schism in the Russian traders' district of medieval Riga called Russische Dorf, which supported those Old Believers who fled from central Russia at the end of the 17th and beginning of the 18th century. Zavoloko's idea was continued by the Soviet ethnographer Antonina Zavarina (1928–2015) in her work on the Russian inhabitants of Eastern Latvia, though she mostly studied the material culture of the Old Believers and their everyday lives (Zavarina 1986).

The first academic scholar of the Latvian Old Believers was Arnold Podmazov (1936–2010), the author of several valuable publications (Podmazov 1970, 1973). Within the framework of the Soviet historiographical tradition, the Old Believers, like any other religious group, were presented in the key to Marxist–Leninist ideology, but it was Podmazov who first investigated the relationship between the Old Believers and the state in the 1920s and 1930s, during the existence of an independent Republic of Latvia. Currently, his research on the Latvian Old Believers has been translated into Latvian and is the only monograph in which this topic is considered in such depth and breadth (Podmazovs 2001).

The Western historiography of the 20th century continued studying church–state relations and the Old Believers trying to oppose the radical views of the Russian Orthodoxy and arguing for a special role of the Old Belief in Russian cultural, political, social, and economic development (Spinka 1941; Miliukov 1942; Casey 1947; Wallace 1961; Blackwell 1965; Cherniavsky 1966). Recent publications on the Old Belief are based largely on materials from Russia and its neighboring territories, analyzing field research data or archival documents obtained by individual scholars in Old Believer communities. Scholars are most interested in the history of the early Old Believer movement and the doctrine or development of the Old Believers after the 1905 revolution, when there was a mass exodus of Old Believers from the territory of the Russian Empire to North and South American territories (Beliajeff 1977; Scheffel 1991a; Michels 1992a, 1992b, 1992c, 2001; Crummey 1993; Robson 1993, 2004, 2014, 2015; Ulianova 1998; Clay 2008; Rogers 2009; Kain 2011; Humphrey 2014; Perrie 2016, 2020; Martin 2018; White 2019). The Russian historiography includes several studies dedicated to the issues of the Russian Orthodox Church and its role in the domestic policy of the empire, including the empire's attitude towards minority religions and the Old Belief, in particular (Polunov 1996; Alekseeva 2000; Firsov 2002; Bendin 2003; Fedorov 2003; Beglov 2014).

In present-day Latvia and the Baltic states, studies on the Old Belief are mostly dedicated to the language, folklore, culture and history of the movement, emphasizing the local peculiarities of a certain community without considering the global problem of church–state relations, although these relations are mentioned as a context for other case studies (e.g., Baranovskij and Potashenko 2005; Potashenko 2006; Ocherki po istorii 2007–2008; Starovery Litvy 2011; Pazuhina 2009, 2011; Pärt 2010; Paert and Schvak 2014; Paert 2016; Korolova and Kovzele 2018, 2019; Korolova et al. 2020; Stasulane 2021).

This paper thus offers the first attempt, based on many studies and historical sources, to consider the issue of marriage among Old Believers in the context of the relationship between church and state. This study includes a long period from the 17th to the 21st century during which the Latvian Old Believers were under the rule of various state regimes; therefore, the considered studies interpret the positions of religion and politics, as well as their impact on the lives of inhabitants, in different ways.

## 3. Materials and Methods

To study the process of developing church, state, and priestless Old Believers' relations and their influence on the question of marital rights, several forms of published and unpublished historical sources have been considered, including archival documents, oral history sources, legislative acts, and periodicals, as well as the data from existing Old Believer parishes collected during oral history expeditions. Many factors influenced the development of ideas about marital rights, most notably: (1) Old Believers' spiritual elders and leaders, who argued about the (im)possibility of marrying without priests and worked

out the ceremony of marriage; (2) state authorities of the Russian Empire, the Republic of Latvia and the USSR, which developed legislation on the registration of marital relations; and (3) parishioners themselves, who decided to follow one or another tactic based on their attitude and needs. Until the beginning of the 20th century, historical sources primarily describe external circumstances; in the data for the 20th century, however, it is possible to find information about the believers themselves and analyze their attitudes towards the issues of marriage, as well as their reflections on the need for religious ceremony as such. The paper traces the mutual influence of internal (the creed) and external (legislation and the general development of society) factors on the process of moving away from religiosity and becoming secularized.

## 4. Findings

### 4.1. Debates on the Acceptability of Marriage and Family Life among Priestless Old Believers

Priestless Old Believers were not able to obtain the sacrament of marriage or conduct the wedding ceremony because they did not have ordained priests. This circumstance led to long and spirited discussions among spiritual fathers and leaders of the Old Belief, who raised questions about marital rights and their legality after the reforms of the Russian Orthodox Church.

The Old Believer Novgorod Council (Sobor) held in 1694 decided to dismiss any possibility of marriage because of the discontinuation of Orthodox priesthood and instead prescribed universal virginity. Married people were therefore unable to join the "true Church" (Smirnov 1898, pp. 176–77). The first generations of the priestless Old Believers were willing to observe celibacy; however, it soon became clear that they could physically die out as a result, and there would be no one to keep the Old Faith. Thus, the discussion was continued by Ivan Alekseyev Starodubsky (1709–1776), who wrote an essay titled "About the Sacrament of Marriage" (1762), in which he criticized the "imaginary" universal celibacy and suggested that Old Believers get married in Orthodox churches (Kruglova 2001, pp. 622–23). This approach was called "Novozhenstvo", and those who accepted it, were known as "Novozheny"[4] (Mal'cev 2006). In the 1750s, this idea was strongly condemned by the Old Believer spiritual leaders, because the Orthodox Church allowed weddings only after conversion into Orthodoxy (Ageeva 2011, p. 190).

In 1752, the Council of Fedoseevcy in Poland decided to employ a different approach to married Old Believers: those who were married by the Orthodox priests before conversion into the Old Belief were called "Starozheny" (literally, "married in an old way") and were allowed religious practice without any restrictions. Those married in Orthodox churches after conversion to the Old Belief, Novozheny, were barred from praying and eating joint meals with other parish members. Such a radical approach was not accepted by the biggest parishes, and by 1777, the Vyg Council ordered the community to allow Novozheny to pray in a certain section of the prayer house[5] and eat at a special table (Juhimenko 2008, p. 12). Furthermore, in the 1780s, a landmark event in accepting the possibility of marriage without priests occurred: an Old Believer spiritual leader from Moscow, Philip (Vasilij Emel'janov 1729–1797), developed a ceremony of blessing for marriage which included reading the canon to the All-Merciful Savior and the Mother of God. Since March 1784, this ceremony has been used for marriage by Moscow Pomorian Old Believers, and it was later accepted by the Vyg Council. Another spiritual leader of the Old Believers from Moscow, Gavriil Skachkov (1745–1821), developed the order of priestless Old Believer marriage, "The canon sung during the wedding", and received permission from the authorities to maintain a "marriage register" that existed till 1837 (Ageeva 2011, p. 192). This kind of marriage became known as a "Moscow Pomorian marriage".

Another denomination of Bespopovcy—Fedoseevcy—continued insisting on "life in virginity" and denying marriages without a priest's blessing because, according to Nomocanon, a marriage without a priest is illegal. Thus, Old Believers of the Baltic region who practiced the Fedoseevcy doctrine had illegal families for a long time; their families were illegal in the eyes of both Christianity and the authorities of the Russian Empire.

After many years of attempting to forcibly baptize Old Believers into Orthodoxy, the state concluded that a special law was needed for marriages among Old Believers.

### 4.2. The Law of 1874

At the beginning of the 19th century, the legislation of the Russian Empire contributed to the establishment of the Orthodox Church as a leader in domestic politics, which, in turn, made other religious groups completely dependent on the will of the emperor. The overall structure of the relationship seemed simple: Russian Orthodoxy had been recognized as the "real" faith of the Russian people, to which as many people of the Russian Empire as possible had to be converted. The attitudes of powerful authorities towards other religions varied from drastically dismissive to acceptable within certain limits. According to the law of the Russian Empire, Russian Orthodoxy was the state religion and dominant Christian confession, with the emperor as its supreme guardian across the country's borders (O prestuplenijah protiv very 1869). Surprisingly, Old Belief—by its nature the closest religion to the Russian Orthodox faith—was perceived as one of the most dangerous enemies of the church, suggesting that the clash was based not only on dogmatic but also political and socioeconomic factors. For centuries, Old Believers faced legal restrictions, humiliation, and even physical punishments; only in the 19th century did missionary activities begin to replace physical coercion (Polunov 2010, p. 254). Still, Old Belief, with more than 2 million followers (Pervaja vseobshhaja perepis' 1899), was an illegal confession, and followers had no inheritance rights for their descendants. To have their children recognized as legitimate, Old Believers used all sorts of tricks. For example, they formally converted to Orthodoxy, got married in an Orthodox church, and received official confirmation of legal marital status and legitimate heirs. Then, the newly converted Orthodox returned to Old Belief religious practice. The authorities turned a blind eye to this tactic because it was advantageous for Orthodox parishes to have a larger number of "official" parishioners, especially among Old Believers, who often paid priests to mark them in the lists of those who had passed confession[6] (Mel'nikov 1909, p. 395).

In the second half of the 19th century, however, the authorities realized the need to officially register the marital status of Old Believers, as the secrecy of the Old Believer parishes and their closed lifestyle made it impossible to collect taxes and get new recruits for compulsory military service. In 1874, the law "On the Rules of Metric Recording of Marriages, Births and Deaths of Dissenters"[7] (hereafter, the Law on Metric Recording) was adopted. The civil registration of Old Believers' families began, but the prohibition on the use of Old Believers' own metric books remained in place[8]. The Law on Metric Recording stated that Old Believers who wished to register their marriage had to announce their intention to local officials orally or in written form. Then, a public notice was posted for seven days. After this term, if there were no oral or written complaints about the forthcoming registration, the pair received a permit to register their marriage (Skorov 1903). To enter a marriage into the register, both partners had to appear in person to the officials and present the permit issued earlier; furthermore, each had to bring two guarantors who would confirm in writing that the marriage was not concluded as a result of a marriage prohibited by the law. In addition, each person was required to present permits issued by parents, guardians, or trustees, and in the case of civil or military service, permits must have been issued by higher management. If military authorities were fierce opponents of the Old Belief, getting a permit could be rather inconvenient or even impossible. However, Old Believer children could be married in an Orthodox church without the permission of their parents or guardians if they converted to Orthodoxy and promised to raise their children within the Orthodox tradition (Skorov 1903). In this way, the existing legal inequalities between the Orthodox and Old Believers affected the formation of official families, creating the conditions to destroy the Old Believers' traditional social model, which held that children could not act without the blessing of their parents. Moreover, Old Believers were against converting to Orthodoxy and sometimes reacted violently (Otchjot Svjato-Vladimirskogo bratstva 1899, p. 9).

There was one more regulation in the Law on Metric Recording concerning Old Believer marriage registration: they had to submit a document that proved the spouses were "schismatics by origin" (*raskol'niki po rozhdeniju*) because those who converted to the Old Belief from other confessions, especially from Orthodoxy, were forbidden to register a marriage. As Old Believers could not marry a person from another confession, according to their religious doctrine, the future spouse had to convert to the Old Belief. Thus, the restriction made registering such a marriage impossible. Several bureaucratic regulations, therefore, prevented Old Believers from registering their marriages in civil metrics and provided an advantage to Orthodox marriages, which were much more accessible and easier to register than an Old Believer marriage. In turn, entering information about children into the register was possible only if there were data from the metric on the registration of the parent's marriage, and only registered children were recognized as legitimate descendants with the right to receive a parental inheritance. In 1891, the Senate of the Russian Empire adopted regulation number 1392, which stated that Old Believer children who were not included in the metrics of Old Believer marriage but were included in the family lists of their fathers were considered adopted and could enjoy the rights of inheritance held by adopted children. In addition, Old Believer marriages that were registered in revision lists before the adoption of the Law on Metric Recording—that is, during the 10th population census (1858–1859)—were considered legal, including children born during those marriages (Skorov 1903).

Thus, the legitimization of the marital rights of Old Believers greatly facilitated inheritance procedures, which particularly affected wealthy plant and factory owners. However, Old Believers who registered in the metrics provided information on their family members and thereby facilitated the acquisition of data by the authorities, who could use it to plan measures against the Old Belief and its followers. Until the legitimization of the Old Belief, the Russian Orthodox Church took measures on "fighting with" the Old Belief and aimed to convert all Old Believers to Orthodoxy (Grizāne 2017).

In 1905, the edict "On strengthening the principles of religious tolerance"[9], also known as the Edict of Tolerance, ended Old Believer family law restrictions. The edict legalized the Old Belief along with other Christian confessions and ended persecution for converting to the Old Belief from Orthodoxy. In addition, Old Believer marriages were recognized. A regulation from 31 January 1907 allowed priestless Old Believer parishes to elect special elders, who had to register Old Believer families in acts of civil status (Vysochajshe utverzhdennoe 31 janvarja 1907 g 1911). A subsequent regulation of the Council of Ministers indicated that Old Believer families and children who were registered in estate family lists (*soslovnye posemejnye spiski*) or other officially recognized documents were considered legal and had inheritance rights (Vysochajshe utverzhdennoe 12 fevralja 1907 g 1911).

Thus, until the legalization of the Old Believers, the issue of marital rights was not resolved. Two processes took place in parallel: on the one hand, the doctrine of family life without the blessing of a priest or the marriage ceremony was developed, and on the other hand, the civil registration of Old Believer families was allowed instead of forced or formal conversion to Orthodoxy.

### 4.3. Old Believer Marital Rights in 1920–1930s

Since the establishment of the independent Republic of Latvia, the position of the Old Belief as a legal confession has been maintained and even strengthened. The Old Believer community obtained a wide range of rights for self-government and founded their managing body, the Central Committee on the Old Believers of Latvia[10] (1920), during the First Congress of the Old Believers in Latvia. Later, the Committee split, and a second institution was founded—the Council of Old Believers' Meetings and Congresses in Latvia[11] (1929). Both institutions registered parishes, distributed government funding, and organized public events and activities. Although Lutheranism became the state religion

of Latvia, the Old Believers' parishes received state support along with other religious organizations. However, there were nuances in the issue of marriage registration.

In 1924, the Latvian government drafted new marriage regulations, which would give Lutheran, Catholic, and Orthodox priests the right to officiate weddings without prior registration in civil institutions. Followers of Jewish, Old Believer, Calvinist, Baptist, and other religious groups, however, had an obligatory civil registration of marital status (Fakul'tativnaja registracija brakov). When this regulation was passed, some religious groups were not satisfied. A Russian newspaper "Segodnja" that depicted opinions of the Russian-speaking inhabitants of Latvia collected data from Old Believers and found out that they were not satisfied with the necessity of civil marriage registration. Under this regulation, new Old Believer families had to pay twice—once to a spiritual leader and once to a civil institution, which was rather expensive for an average peasant. Moreover, Old Believer spiritual leaders had their own registration books for marriage acts (Staroobrjadcy o registracii brakov). It is important to note that on 17 October 1906, the Supreme decree on the procedure for the formation and operation of Old Believer parishes[12] gave the right to Old Believer parishes to register marriages: a parish spiritual leader simply had to send data on the marital status of his parishioners to civil institutions. A similar decision regarding the Old Believers of Latvia was made in March 1928: all religious groups, including Old Believers, had the right to register marriages according to their religious traditions and without registration in civil registry offices (Laulību likuma). The additional regulations also defined the duty of spiritual leaders to provide data on marriage acts to state institutions within 14 days (Pārgrozījumi un papildinājumi likumā par laulību). According to this law, the issues of Old Believers' marital rights were transferred to the Old Believers themselves. However, due to a lack of general education and poor knowledge of the Latvian language (Grizāne 2021), Old Believer spiritual leaders did not meet the task: there were several cases of litigation concerning incorrectly completed documents and resulting problems with employment, inheritance rights and so on. This problem was solved by allowing Old Believers to hire Latvian-speaking secretaries (Kirņičanska and Saleniece 2022).

The state-controlled registration of the marital status of Old Believers stimulated the spread of civil divorces. Traditionally, Old Believer families were very stable, and divorces were prohibited. In exceptional cases, however, divorces could be approved by the spiritual leader of the parish, who knew about the family life and health of his parishioners; some spiritual leaders even consulted with doctors to clarify information about certain diseases before making decisions about divorces (Vārdi un darbi). Latvian periodicals mention more than 20 cases of Old Believer divorces from 1924 to 1936, primarily in the Riga and Daugavpils regions[13]. All of these divorces were initiated by women, and all court decisions were made without the presence of the husbands at the court sessions. The newspapers did not mention the reasons for divorce, but it is known that all ex-wives kept their children and received some monetary compensation. Nonetheless, the question remains: were the divorce cases coordinated with the Old Believer spiritual leaders, or did the women use the state legislative system to terminate unwanted family ties, bypassing the spiritual leader's decision? Civil divorces, unlike marriages, were controlled by civil institutions only, so we can argue that this situation stimulated the secularization of the Old Believer community and made parishioners more independent from their spiritual leaders. In addition, civil divorces probably improved some women's conditions since Old Believer families were traditionally very patriarchal, and the choice of a future spouse, as a rule, was controlled by one's parents. While spiritual leaders usually paid attention to illnesses such as mental disorders and infertility in the case of divorce, the psychological problems of family members were not considered a priority.

On the doctrinal level, Latvian Old Believer parishes continued the process of transitioning from Fedoseevcy teachings to Pomorian, as they also gradually accepted family life and the marriage ceremony preceding it. A researcher of the Old Belief in Latvia, Arnold Podmazov (1936–2010), noted that despite the constant debate about whether married

and celibate followers could pray together, the presence of common central government bodies, the publication of joint religious literature and the permission for spiritual leaders to transition from one doctrine to another contributed to the consolidation of parishes, with the subsequent transition to the Pomorian doctrine (Podmazov 1970, pp. 63–64). The Old Believers' Spiritual Committee played an active role in the rapprochement of Fedoseevcy and Pomorian groups and the solution of marriage issues. In 1926, the Committee declared that those who were not blood relatives could be married after receiving permission from their parents and a spiritual leader (Nikonov 2008, p. 188). On the eve of the Second World War, the Latvian Old Believers overwhelmingly accepted the Pomorian doctrine and have been considered followers of the Pomorian doctrine ever since. By the 1960s, there were only a few Fedoseevcy parishes in the eastern part of Latvia (Podmazov 1970, p. 64).

Along with the marriage ceremony, which was conducted by spiritual leaders in prayer houses, the Old Believers of Latvia retained some elements of older marriage rituals that were probably common in pre-Christian times. There is evidence that in the 1920–1930s, the custom of stealing brides was still practiced: during the annual autumn fairs, young men, usually conspiring in advance, took young women away, and then came to the woman's parents' house to decide whether there would be a marriage ceremony. Such cases were rather few; instead, traditionally, potential spouses and their parents concluded a kind of agreement (*sgovor*) and then organized a marriage ceremony with the following celebrations (Korolova et al. 2021).

During the interwar period, most Old Believer parishes reconciled on the issue of marriage relations, and the acceptance of Pomorian teaching, and cooperation with the state also made it possible to consolidate Old Believer traditions of marriage. After several decades of cultural prosperity and development, the Old Believers' religious life was influenced by the Soviet regime and subsequent implementation of an atheistic worldview and a state ban on the official registration of family relations through religious ceremonies.

*4.4. Forced Secularization in the Soviet Period and the Present Day*

The new Soviet government initially took a rather neutral attitude towards the many Old Believer parishes operating in the annexed territories. In turn, the Baltic Old Believers emphasized support for "the policy of the Soviet government for the common good of our native Fatherland" (Beliakova 2014, p. 247) in their letters and documents, and they even declared the Soviet regime was the "God-established authority" for which it was prescribed to pray on public holidays (Staroobrjadcheskij cerkovnyj kalendar' 1949, p. 78). In 1945, the Council for Religious Cults[14] intended to create centralized management structures for religious denominations or reorganize the existing ones. In the late 1940s, it became clear that the creation of a central administration for the Old Believers of the Pomorian denomination had no strategic significance for the Soviet government. This process was curtailed due to the radically changed course of state policy. The desire of the leadership of the Lithuanian Old Believer Council to use the short-term interest of the Soviet government to strengthen the Old Believer movement was categorically rebuffed, suspended, and severely punished (Beliakova 2014, p. 262). Some researchers of the history of the Baltic region note that, in 1944–1953, the Soviet government transferred the experience of harsh anti-religious policy to the newly annexed territories rather than creating new, regional models of state–church relations based on a more pragmatic approach (Potashenko 2022, p. 278). In the course of the mass deportations of 1941 and 1949, thousands of people were forced to leave their native lands, including Old Believer spiritual leaders and parishioners (Saleniece 2008). Declericalization and the arrests of clergymen were widespread practices (Kiope et al. 2020, pp. 143–44; Krūmiņa-Koņkova 2015, pp. 149–50). However, Old Believer parishes continued their activities and, along with the liberalization of the Soviet regime in the 1950s and 1960s, some Old Believer spiritual leaders expressed the idea that Old Believers "had never been free like now" (Podmazov 1970, pp. 98–99). After the Old Believer Council of the USSR Pomorian parishes in Vilnius in October 1966, the Lithuanian Old Believer Council retained its role as the unifying body for the Baltic and Moscow Old

Believer parishes. The Council maintained records of Old Believer parishes and formally approved new spiritual leaders, as well as organized the so-called spiritual courts to sort out disputes concerning doctrine (Podmazovs 2001, pp. 143–44).

Introducing atheistic propaganda, the Soviet authorities insisted on replacing religious traditions with secular ones: Old Believer marriage ceremonies in a prayer house were replaced with the "Komsomol" or Young Communist League members' marriage at a state registry office. The legislation of the USSR stated that marriages registered by religious institutions were illegal and did not imply that children had inheritance rights. From a legal point of view, Old Believers had to repeat the 19th century experience of confirming marital status through a civil registration procedure (Korolova et al. 2021, p. 169). In most cases, marriage ceremonies were reduced to blessings by parents and/or a spiritual leader outside a prayer house. Usually, this happened at home, hidden from the authorities and those who could inform the authorities about it. Old Believers, especially those who held public positions, had to abandon their religious practice or preserve it in secret because of the threat of being punished with public disgust and unemployment (Grizāne 2022). In this way, the Old Believers of Latvia, along with other religious groups, had to survive under atheistic propaganda. According to sociological observations from the end of the 1960s in Eastern Latvia, Old Believers' adherence to the old rites weakened, but some "cult elements" persisted steadily, and the tradition of marriage ceremonies has been preserved (Podmazov 1970, p. 105).

Currently, according to the legislation of the restored Republic of Latvia, the church is separated from the state; everybody is equal regardless of their attitude towards religion (Law on Religious Organisations 1995). The Old Believers of Latvia are unified by the Pomorian Old Orthodox Church of Latvia and the Central Council of the Church as a managing body. The last Fedoseevcy parish, consisting of four people, exists separately (Nesterenko 2017). According to the Law on the Pomorian Old-Orthodox Church of Latvia, Old Believers are allowed to perform marriage ceremonies in accordance with the procedures laid down in their religious ceremonies (Law on the Pomorian Old-Orthodox Church of Latvia 2007). Nonetheless, the number of marriages concluded in Old Believer prayer houses is rather low: from 13 unions in 2005 and 2008 to no ceremonies from 2018 to 2021 (Publiskais pārskats 2006–2022). Considering the number of Old Believers—around 60,000—this number is very small. This situation may result from the consequences of Soviet secularization and the global crisis of Christianity or the spread of ideas about diversity and tolerance for various manifestations of human culture and worldviews. There has been a significant increase in the number of interfaith marriages, which was strictly prohibited in the Old Belief until quite recently; these marriages are concluded in churches of other confessions or through the registration of acts of civil status. Finally, the lack of Old Believer marriages could be due to changing attitudes towards the institution of the family, which is no longer seen as a God-blessed union of a man and a woman for life but a civil marriage or cohabitation, which is easy to terminate. Being free in their choices, most Old Believers prefer not to delve into the specifics of the creed and instead live according to modern ideas about the family.

## 5. Conclusions

Old Believers and religious freedom have long been a topic of interest. Initially, authors from the Russian Orthodox Church represented the Old Belief as a dangerous heretical sect that had to be defeated, and its followers had to be "returned" to the Orthodoxy, while Western researchers considered the Old Belief an important social movement that played a crucial role in the development of Russian culture. Modern ideas about diversity, including religious diversity, highlight the need to consider any phenomenon from different points of view. The history of the Old Believers, which is very extensive and diverse, is one of the most complex topics regarding church–state relations and their influence on minority religious groups.

As a religious movement, the Old Belief has come a long way, from a number of illegal sects to denominations with developed doctrines. By rejecting the reformed Orthodoxy and the church hierarchy, the Old Believers deprived themselves of the opportunity to perform the sacrament of marriage—one of the sacraments that only an ordained priest can perform. The state and the Russian Orthodox Church perceived this choice as heretical and took various measures to forcibly return the Old Believers to Orthodoxy, which was ultimately fruitless. The families of Old Believers, considered illegal by the state and the Orthodox Church, continued to follow their faith.

At first, the Old Believers of Latvia followed the teaching of the Fedoseevcy denomination and accordingly stood against any family life. The question of marital rights was a topic of debate for a long time: Old Believers held onto the tenet of not having family ties because they could not imagine a true Christian family without the sacrament of marriage. Nevertheless, at the turn of the 20th century, Latvian Fedoseevcy parishes decided to adopt the marriage ceremony from the community of the Pomorian doctrine. This process was also stimulated by the 1874 law on the civil registration of marriages for Old Believers, the subsequent legalization of the Old Belief, and the adoption of the 1906 law allowing Old Believers' spiritual leaders to register marriages themselves. This trend was continued by the government of the Republic of Latvia, which developed a procedure for registering marriages that allowed Old Believers to avoid additional registration at the registry office. State support contributed to the consolidation of the Old Believers' parishes of Latvia, on the one hand, and, on the other hand, facilitated the emergence of civil divorces.

Under the influence of the Soviet regime's atheistic propaganda, the Old Believer community of Latvia was again divided into separate parishes, and each person faced a personal choice: abandon religiosity or continue to keep traditions. As a result, the civil procedure of marriage registration has again become dominant for the legalization of family relations. Old Believers, officially called Old Orthodox, consider themselves very conservative and aim to preserve the old traditions of Orthodoxy in the face of problems such as aging parishioners and the reduction of young followers. Having the right to legal marriage ceremonies, most Old Believers choose civil registration or wedding according to traditions of other confessions. Even a conservative and initially closed group such as Old Believers cannot completely withstand the challenges of modernity and avoid the influence of mainstream processes.

**Funding:** This research received no external funding.

**Data Availability Statement:** The data from this study is not publicly available.

**Conflicts of Interest:** The author declares no conflict of interest.

## Notes

[1]  In time the Old Belief divided into two main groups: (1) Popovcy (priestly), who accepted Orthodox priests but followed the pre-reformed church service, and (2) Bespopovcy (priestless), who denied any priesthood and instead elected spiritual leaders to perform the main rituals and sacraments that lay people were allowed to perform (baptism, confession, and communion).

[2]  This explains the Russian name of the Old Believers, *staroobrjadcy*, where "starii" means "old" and "obrjad" means "ritual". At the end of the 19th century, this name was replaced by the designation "schismatics" (*raskolniki*), which was used in the official documents and legislation of the Russian Empire. This approach is also widespread in Western historiography (e.g., Scheffel 1991b). In the present day, Old Believers prefer the name Old Orthodox, *starovery,* or *staroverci*, where "vera" means "belief", which highlights their commitment to the pre-reformed Orthodoxy.

[3]  Bespopovcy—literally "without a pop [priest]"—denied Orthodox priesthood because it accepted the reforms of Patriarch Nikon in the 1660s and lost its divine blessing. The biggest Bespopovcy denominations—Pomorians (or Pomorcy) and Fedoseevcy—differ in their attitude towards marriage. On the contrary, Popovcy accepted Orthodox priests but followed the pre-reformed religious practice.

[4]  "Novozheny" literally means "married in a new way".

[5]  Priestless Old Believers cannot have altars at churches, which is why they call their places for worship "prayer houses" (*mollenaja*).

6    It was obligatory for all Orthodox followers to pass confession; otherwise, they might be considered "dissenters" or sectarians (Mironov 2007). Old Believers could not afford this, because that would mean the betrayal of the Old Belief, which is why they paid priests not to be punished for breaching rules.

7    *O pravilah metricheskoj zapisi brakov, rozhdenija i smerti raskol'nikov.*

8    The secular metric was introduced in Russia for the first time by Peter I the Great in 1702 in Moscow Orthodox parishes. After 1722, it was widespread across the Russian Empire. In 1802, the Holy Synod passed the rules "About the Content of Metric Books in the Prescribed Order" (*O soderzhanii v predpisannom porjadke metricheskih knig*), which forbade keeping the books in priests' houses (Novosel'skij 1916, p. 42).

9    *Ob ukreplenii nachal veroterpimosti.*

10    *Central'nyj komitet po delam staroobrjadcev Latvii.*

11    *Sovet staroobrjadcheskih soborov I s"ezdov v Latvii.*

12    The full name of the decree is "On the procedure for the formation and operation of Old Believer and sectarian parishes and on the rights and obligations of the members of the parishes of Old Believer denominations and sectarians separated from Orthodoxy" (*O porjadke obrazovanija i dejstvija staroobrjadcheskih i sektantskih obshhin i o pravah i obazannostjah vhodjashhih v sostav obshhin posledovatelej staroobrjadcheskih soglasij i otdelivshihsja ot pravoslavija sektantov*).

13    The cases of Old Believer divorces are found in the periodicals: Rīgas apgabaltiesas 1. civilnodaļa. *Valdības Vēstnesis 04.01.1924*; Rīgas apgabaltiesas 1. civilnodaļa. *Valdības Vēstnesis 28.03.1924*; Rīgas apgabaltiesas 4. civilnodaļa. *Valdības Vēstnesis 24.12.1924*; Rīgas apgabaltiesas 4. civilnodaļa. *Valdības Vēstnesis 24.08.1925*; Rīgas apgabaltiesas 4. civilnodaļa. *Valdības Vēstnesis 24.02.1928*; Rīgas apgabaltiesas 4. civilnodaļa. *Valdības Vēstnesis 22.01.1929*; Rīgas apgabaltiesas 4. civilnodaļa. *Valdības Vēstnesis 23.02.1929*; Rīgas apgabaltiesas 4. civilnodaļa. *Valdības Vēstnesis 06.03.1929*; Rīgas apgabaltiesas 4. civilnodaļa. *Valdības Vēstnesis 25.05.1929*; Rīgas apgabaltiesas 4. civilnodaļa. *Valdības Vēstnesis 18.10.1929*; Rīgas apgabaltiesas 4. civilnodaļa. *Valdības Vēstnesis 20.01.1930*; Rīgas apgabaltiesas 4. civilnodaļa. *Valdības Vēstnesis 21.04.1931*; Rīgas apgabaltiesas 4. civilnodaļa. *Valdības Vēstnesis 06.07.1932*; Rīgas apgabaltiesas 4. civilnodaļa. *Valdības Vēstnesis 04.10.1932*; Rīgas apgabaltiesas 4. civilnodaļa. *Valdības Vēstnesis 06.04.1934*; Rīgas apgabaltiesas 4. civilnodaļa. *Valdības Vēstnesis 16.05.1935*; Rīgas apgabaltiesas 4. civilnodaļa. *Valdības Vēstnesis 11.12.1935*; Latgales apgabaltiesas 1. civilnodaļa. *Valdības Vēstnesis 27.02.1930*; Latgales apgabaltiesas 1. civilnodaļa. *Valdības Vēstnesis 22.03.1934*; Daugavpils apgabaltiesas 3. civilnodaļa. *Valdības Vēstnesis 09.01.1936*; Daugavpils apgabaltiesas 1. civilnodaļa. *Valdības Vēstnesis 28.12.1936*.

14    *Sovet po delam religioznyh kul'tov.*

# References

## Primary Sources

Die Wirksamkeit des heiligen Synods im Jahre 1874. *Rigasche Zeitung*, nr. 69, 24 March 1874.

Iekšzemes atskats. Par raskolnikiem Baltijas guberņās. Rota, nr. 1, 1 January 1886.

Iekšzemes ziņas. Balss, nr. 1, 1 January 1886.

Moskau. Rigasche Zeitung, nr. 53, 5 March 1869.

Sv. sinoda virsprokurora vispadevīgā vēstījuma. Mājas Viesis, nr. 43, 25 October 1886.

Daugavpils apgabaltiesas 1. civilnodaļa. Valdības Vēstnesis 28 December 1936.

Daugavpils apgabaltiesas 3. civilnodaļa. Valdības Vēstnesis 9 January 1936.

Fakul'tativnaja registracija brakov? Segodnja, nr. 223, 1 October 1924.

Latgales apgabaltiesas 1. civilnodaļa. Valdības Vēstnesis, nr. 66, 22 March 1934.

Latgales apgabaltiesas 1. civilnodaļa. Valdības Vēstnesis, nr. 48, 27 February 1930.

Laulību likuma. Latvijas Kareivis, nr. 55, 8 March 1928.

Pārgrozījumi un papildinājumi likumā par laulību. Latvijas Kareivis, nr. 54, 7 March 1928.

Rīgas apgabaltiesas 1. civilnodaļa. Valdības Vēstnesis, nr. 3, 4 January 1924.

Rīgas apgabaltiesas 1. civilnodaļa. Valdības Vēstnesis, nr. 73, 28 March 1924.

Rīgas apgabaltiesas 4. civilnodaļa. Valdības Vēstnesis, nr. 224, 4 October 1932.

Rīgas apgabaltiesas 4. civilnodaļa. Valdības Vēstnesis, nr. 53, 6 March 1929.

Rīgas apgabaltiesas 4. civilnodaļa. Valdības Vēstnesis, nr. 75, 6 April 1924.

Rīgas apgabaltiesas 4. civilnodaļa. Valdības Vēstnesis, nr. 147, 6 July 1932.

Rīgas apgabaltiesas 4. civilnodaļa. Valdības Vēstnesis, nr. 282, 11 December 1935.

Rīgas apgabaltiesas 4. civilnodaļa. Valdības Vēstnesis, nr. 109, 16 May 1935.

Rīgas apgabaltiesas 4. civilnodaļa. Valdības Vēstnesis, nr. 237, 18 October 1929.

Rīgas apgabaltiesas 4. civilnodaļa. Valdības Vēstnesis, nr. 15, 20 January 1930.

Rīgas apgabaltiesas 4. civilnodaļa. Valdības Vēstnesis, nr. 87, 21 April 1931.

Rīgas apgabaltiesas 4. civilnodaļa. Valdības Vēstnesis, nr. 18, 22 January 1929.

Rīgas apgabaltiesas 4. civilnodaļa. Valdības Vēstnesis, nr. 44, 23 February 1929.

Rīgas apgabaltiesas 4. civilnodaļa. Valdības Vēstnesis, nr. 45, 24 February 1928.

Rīgas apgabaltiesas 4. civilnodaļa. Valdības Vēstnesis, nr. 187, 24 August 1925.

Rīgas apgabaltiesas 4. civilnodaļa. Valdības Vēstnesis, nr. 293, 24 December 1924.

Rīgas apgabaltiesas 4. civilnodaļa. Valdības Vēstnesis, nr. 114, 25 May 1929.

Alksnis J. Vārdi un darbi. Brīvā Zeme, nr. 114, 21 May 1938.

Kallistratov. Staroobrjadcy o registracii brakov. Segodnja, nr. 27, 4 February 1925.

## Secondary Sources

Ageeva, Elena. 2011. Staroobrjadcheskij polemicheskij sbornik XIX veka: k voprosu o brake u staroobrjadcev. *Vestnik cerkovnoj istorii* 3–4: 191–218.

Alekseeva, Svetlana. 2000. Institut sinodal'noj Ober-Prokuratury i ober prokurory Svjatejshego Sinoda v 1856–1904 gg. *Nestor* 1: 292–310.

Baranovskij, Vasilij, and Grigorij Potashenko. 2005. *Staroverie Baltii i Pol'shi: Kratkij istoricheskij i biograficheskij slovar'*. Vil'njus: Aidai.

Beglov, Alexey. 2014. Konformizm prihodskoj reformy K. P. Pobedonosceva. *Quaestio Rossica* 3: 107–23. [CrossRef]

Beliajeff, Anton. 1977. The Old Believers in the United States. *The Russian Review* 36: 76–80. [CrossRef]

Beliakova, Nadezhda. 2014. O popytke sozdanija organizacionnoj struktury u staroobrjadcevbespopovcev Pribaltiki v 1940e gg. In *Latvijskie starovery:istoricheskij opyt sohranenija identichnosti. Sbornik statej*. Edited by N. Pazuhina. Riga: Latvijas Universitātes aģentūra LU Filozoijas un socioloģijas institūts, pp. 247–62.

Bendin, Aleksandr. 2003. Veroterpimost' i problemy nacional'noj politiki Rossijskoj imperii (vtoraja polovina XIX—nachalo XX v.). *Minskie eparhial'nye vedomosti* 4: 52–57.

Blackwell, William. 1965. The Old Believers and the Rise of Private Industrial Enterprise in Early Nineteenth-Century Moscow. *Slavic Review* 24: 407–24. [CrossRef]

Casey, Robert. 1947. The Cultural Mission of Russian Orthodoxy. *The Harvard Theological Review* 40: 257–75. [CrossRef]

Cherniavsky, Michael. 1966. The Old Believers and the New Religion. *Slavic Review* 25: 1–39. [CrossRef]

Clay, Eugene. 2008. Mapping the Limits of Orthodoxy: Russian Orthodoxy Missionary in Perm' Diocese, 1828–1912. *Russian History* 35: 113–28. [CrossRef]

Crummey, Robert. 1993. Old Belief as Popular Religion: New Approaches. *Slavic Review* 52: 700–12. [CrossRef]

Farmakovskij, Vladimir. 1866. O protivogosudarstvennom jelemente v raskole. *Otechestvennye zapiski* 12: 488–518.

Fedorov, Vladimir. 2003. *Russkaja Pravoslavnaja Cerkov' i gosudarstvo. Sinodal'nyj period. 1700–1917*. Moskva: Izd. Russkaja Panorama.

Firsov, Sergej. 2002. *Russkaja Cerkov' nakanune peremen (konec 1890-h–1918 gg.)*. Moskva: Duhovnaja biblioteka.

Franck, Adolphe. 1859. *Le raskol. Essai sur les sectes religieuses en Russie*. Paris: Libraire-Editeur A. Franck.

Gagarin, Father Jean. 1872. *The Russian Clergy*. Translated by Charles Du Gard Makepeace. London: Burns and Oates.

Grizāne, Maija. 2017. Pareizticīgo misionāru darbība vecticībnieku vidū Vitebskas guberņas Rēzeknes, Daugavpils un Ludzas apriņkos (1894–1901). *Reliģiski-filozofiski raksti* XXII: 142–63. [CrossRef]

Grizāne, Maija. 2021. Soviet Schools and Old Believers' Children's Religiosity in Eastern Latvia. *Reliģiski-filozofiski raksti* XXXI: 302–32. [CrossRef]

Grizāne, Maija. 2022. Soviet Secularisation: The Experience of the Old Believers in Eastern Latvia. *Historická Sociologie* 14: 23–35. [CrossRef]

Gromoglasov, Il'ja. 1895. *O sushhnosti i prichinah russkogo raskola tak nazyvaemogo staroobrjadstva*. Sergiev Posad: 2 tipografija A. I. Snegirevoj.

Heard, Albert. 1887. *The Russian Church and Russian Dissent, Comprising Orthodoxy, Dissent, and Erratic Sects*. New York: Harper & Brothers.

Humphrey, Caroline. 2014. Schism, Event, and Revolution: The Old Believers of Trans-Baikalia. *Current Anthropology* 55: 216–25. [CrossRef]

Ivanovskij, Nikolaj. 1883. *Kriticheskij razbor uchenija nepriemljushhih svjashhenstva staroobrjadcev o cerkvi i tainstvah*. Kazan': Tipografija Imperatorskogo Universiteta.

Iz praktiki svjashhennika-missionera. 1900. *Pribavlenie k Chernigovskim eparhial'nym izvestijam* 6: 203–9.

Juhimenko, Elena. 2008. *Pomorskoe staroverie v Moskve i hram v Tokmakovom pereulke*. Moskva: VINITI.

Kain, Kevin. 2011. Reading between the (Confessional) Lines: The Intersection of Old Believer Manuscript Books and Images with Print Cultures of Late Imperial Russia. In *The Space of the Book: Print Culture in the Russian Social Imagination*. Edited by M. Remnek. Toronto: University of Toronto Press, pp. 165–200.

Kak prezhde obrashhali raskol'nikov v Pravoslavie domashnimi sredstvami. 1882. *Istoricheskij vestnik* 8: 473–75.

Kiope, Māra, Inese Runce, and Anita Stasulane. 2020. Trajectories of Atheism and Secularisation in Latvia: From the German Enlightenment to Contemporary Secularity. In *Freethought and Atheism in Central and Eastern Europe*. Edited by Tomáš Bubík, Atko Remmel and David Václavík. New York: Routledge, pp. 137–54.

Kirņičanska, Ina, and Irēna Saleniece. 2022. Old Believers' identity and religious freedom literacy: The case of Latvia from 1918 to the present days. Paper presented at the Religion between Governance and Freedom, University of Padova, Padova, Italia, September 22–23.

Korolova, Jelena, and Oksana Kovzele. 2018. Dual Constractions of Concrete Nouns as a Projection of Latgalian Old Believers Language Reflections. Paper presented at 5th International Multidisciplinary Scientific Conference on Social Sciences and Arts SGEM 2018, Albena, Bulgaria, August 26–September 1; pp. 187–94. [CrossRef]

Korolova, Jelena, and Oksana Kovzele. 2019. Linguistic peculiarities of invectives in dialect society (Latgalian Old Believers). Paper presented at 6th SWS International Scientific Conference on Arts and Humanities, Albena, Bulgaria, August 26–September 1; pp. 705–12. [CrossRef]

Korolova, Jelena, Ilze Kacane, and Oksana Kovzele. 2020. Ethno-religious minority in the borderland: The case of Latgalian Old Believers (Latvia). *Trames* 24: 519–31. [CrossRef]

Korolova, Jelena, Oksana Kovzele, Ilze Kacane, and Maija Grizane. 2021. Transformations of Old Believer Wedding Rites in Latvia: The Case of Latgale. *Journal of Ethnology and Folkloristics* 15: 159–78. [CrossRef]

*Kratkaja istorija raskola. O bespopovcah*. 1866. Moskva: tipografija "Russkih vedomostej".

Kruglova, Tat'jana. 2001. Alekseev Ivan. In *Pravoslavnaja jenciklopedija. T. 1*. Moskva: Cerkovno-nauchnyj centr RPC «Pravoslavnaja jenciklopedija», pp. 622–23.

Krūmiņa-Koņkova, Solveiga. 2015. Sadarbība starp LPSR drošības iestādēm un PSRS Reliģisko kultu lietu padomes pilnvaroto LPSR (1944–1954). In *VDK Zinātniskās izpētes Komisijas raksti: 1. Sējums. Totalitārisma sabiedrības kontrole un represijas*. Rīga: LPSR Valsts drošības komitejas zinātniskās izpētes komisija, pp. 147–93.

Law on Religious Organisations. 1995. Legal Acts of the Republic of Latvia. Available online: https://likumi.lv/ta/en/en/id/36874 (accessed on 29 May 2023).

Law on the Pomorian Old-Orthodox Church of Latvia. 2007. Laws and Regulations of the Republic of Latvia (in English). State Language Centre. Available online: https://www.vvc.gov.lv/en/laws-and-regulations-republic-latvia-english/law-pomorian-old-orthodox-church-latvia (accessed on 29 May 2023).

Mal'cev, Aleksandr. 2006. *Staroobrjadcheskie bespopovskie soglasija v XVIII–nachale XIX v. (problema vzaimootnoshenij)*. Edited by Nikolaj Pokrovskij and Natal'ja Gur'janova. Novosibirsk: ID "Sova".

Martin, Dominic. 2018. Différances of Doverie: (Mis)trust and the Old Faith in the Russian Far East. In *Trust and Mistrust in the Economies of the China-Russia Borderlands*. Edited by C. Humphrey. Amsterdam: Amsterdam University Press, pp. 143–78.

Mel'nikov, Pavel. 1909. Schislenie raskol'nikov. In *Polnoe sobranie sochinenij P.I. Mel'nikova (Andreja Pecherskogo), izd. 2, tom 7*. Sankt-Peterburg: Izdanie T-va A. F. Marks, pp. 384–409.

Michels, Georg. 1992a. The First Old Believers in Ukraine: Observations about Their Social Profile and Behavior. *Harvard Ukrainian Studies* 16: 289–313.

Michels, Georg. 1992b. The Solovki Uprising: Religion in Northern Russia. *The Russia Review* 51: 1–15. [CrossRef]

Michels, Georg. 1992c. The Violent Old Belief: An Examination of Religious Dissent on the Karelian Frontier. *Russian History* 19: 203–29. [CrossRef]

Michels, Georg. 2001. Rescuing the Orthodox: The Church Policies of Archbishop Afanasii of Kholmogory, 1682–1702. In *Of Religion and Empire: Missions, Conversion, and Tolerance in Tsarist Russia*. Edited by Robert P. Geraci and Michael Khodarkovsky. Ithaca and London: Cornell University Press, pp. 19–37.

Miliukov, Paul. 1942. *Outlines of Russian Culture, Part 1: Religion and the Church*. Edited by M. Karpovich. Philadelphia: University of Pennsylvania Press.

Miller, Edward Waite. 1907. Some Distinctive Features of Russian Christianity. *The American Journal of Theology* 11: 597–623. [CrossRef]

Mironov, Boris. 2007. Ispovednyj i metricheskij uchet v imperskoj Rossii. In *Materialy cerkovno-prihodskogo ucheta naselenija kak istoriko-demograficheskij istochnik: Sbornik statej*. Edited by Vladimir Vladimirov. Barnaul: Izdatelstvo AltGU, pp. 8–47.

*Missionerskoe obozrenie*. 1896–1908. Kiev and Sankt-Peterburg: Tipografija I. I. Chokolova; Tipo-litografija V. V. Komarova; Ju. A. Skvorcova; V. M. Skvorcov.

Mouravieff, Andrew. 1842. *History of the Church of Russia*. Translated by R. W. Blackmore. Oxford: John Henry Parker.

Nesterenko, Kristina. 2017. Transformation of the identity of Old Believers in the Baltic states (Latvia and Estonia) and Romania. *Kulturní studia* 1: 62–78. [CrossRef]

Nikiforovskij, Ivan. 1892. *Osnovnaja osobennost' staroobrjadcheskogo raskola*. Samara: Tipo-litografija N. A. Zhdanova.

Nikonov, Vladimir. 2008. *Staroverie Latgalii: Ocherki po istorii starovercheskih obshhestv Rezhickogo i Ljucinskogo uezdov (2-ja polovina XVII—1-ja polovina XX vv.)*. Rezekne: Rezeknenskaja kladbishhenskaja staroobrjadcheskaja obshhina.

Novgorodskij, Pavel. 1888. *Svet vo t'me raskola. zamechatel'nye sluchai obrashhenija raskol'nikov v Pravoslavie. Chast' 1–2*. Vladimir: Lito-Tipografija P. F. Novgorodskogo.

Novosel'skij, Sergej. 1916. *Smertnost' i prodolzhitel'nost' zhizni v Rossii*. Petrograd: Tipografija Ministerstva vnutrennih del.

O prestuplenijah protiv very, i o narushenii ograzhdajushhih onuju postanovlenija. 1869, In *Ulozhenie o nakazanijah ugolovnyh i ispravitel'nyh*. Sankt-Peterburg: Razdel II, pp. 52–69.

*Obzor Vitebskoj gubernii za 1894 god*. 1894. Vitebsk: Gubernskaja Tipografija.

*Ocherki po istorii i kul'ture staroverov Jestonii. ch. 1, ch. 2*. 2007–2008. Humaniora: Lingua Russica. Tartu: Izdatel'stvo Tartuskogo Universiteta.

Otchjot Svjato-Vladimirskogo bratstva za 1898 God. 1899. Collection 2556 "Vitebskoe eparhial'noe Vladimirskoe bratstvo", inventory list 1, folder 1. National Historical Archives of Belarus, Minsk, Belarus.

Paert, Irina. 2016. Memory of Socialism and the Russian Orthodox Believers in Estonia. *Journal of Baltic Studies* 47: 497–512. [CrossRef]

Paert, Irina, and Toomas Schvak. 2014. Orthodox Education in the Baltic Provinces of Imperial Russia and Independent Estonia from 1840s till 1941. *Quaestio Rossica* 2: 142–58. [CrossRef]

Palmer, William. 1871. *The Patriarch and the Tsar: The Replies of Nikon*. Translated by W. Palmer. London.

*Pamjatnaja knizhka Vitebskoj gubernii za [1890–1905] god*. 1891–1906. Vitebsk.

Pärt, Irina. 2010. *Spiritual Elders*. DeKalb: Northern Illinois University Press.

Pazuhina, Nadezhda. 2006. Pagātne kā nākotne: Vecticībnieku kultūras pieredze. *Kultūras krustpunkti* 2: 206–12. [CrossRef]

Pazuhina, Nadezhda. 2009. *Kulturpraktiken der russisch-orthodoxen Altgläubigen Lettlands. Erfahrungen von Stabilität und Wandel in priesterlosen Gemeinschaften (1920–1939 und 1991–2006)*. Saarbrücken: Südwestdeutscher Verlag für Hochschulschriften.

Pazuhina, Nadezhda. 2011. The "Native-History" (Rodnaya Starina) Through the Glimpse of Latvian Old Believers: Problems of Cultural Identity in Poly-cultural Milieu. In *National Minorities, Identity, Education*. Praha: Institute of Contemporary History of the Academy of Sciences of the Czech Republic, pp. 77–89.

Perrie, Maureen. 2016. The Old Believers and Praying for the Tsar in Seventeenth-Century Russia. *The Slavonic and East European Review* 94: 243–58. [CrossRef]

Perrie, Maureen. 2020. In Search of an Apostolic Succession: Russian Old Believers and the Legend of Belovod'e. *The Slavonic and East European Review* 98: 266–97. [CrossRef]

*Pervaja vseobshhaja perepis' naselenija Rossijskoj imperii, 1897 g*. 1899. Sankt-Peterburg: Izdanie Central'nogo statisticheskogo komiteta Ministerstva vnutrennih del pod redakciej N. A. Trojnickogo.

Podmazov, Arnold. 1970. *Staroobrjadchestvo v Latvii*. Riga: Izdatel'stvo "Liesma".

Podmazov, Arnold. 1973. *Cerkov' bez svjashhenstva*. Riga: Liesma.

Podmazovs, Arnolds. 2001. *Vecticība Latvijā*. Rīga: LU Filozofijas un socioloģijas institūts.

Polunov, Aleksandr. 1996. *Pod vlast'ju ober-prokurora. Gosudarstvo i cerkov' v jepohu Aleksandra III*. Moskva: AIRO-XX.

Polunov, Aleksandr. 2010. *K. P. Pobedonoscev v obshhestvenno-politicheskoj i duhovnoj zhizni Rossii*. Edited by M. Ajlamazjan. Moskva: Rossijskaja politicheskaja jenciklopedija (ROSSPJeN).

Potashenko, Grigorij. 2006. *Staroverie v Litve: Vtoraja polovina XVII—nachalo XIX vv.: Issledovanija, dokumenty i materialy*. Vil'njus: Aidai.

Potashenko, Grigorij. 2022. Staroverie Litvy v 1944–1953 gg.: «Novyj kurs» sovetskij vlastej, dejatel'nost' VSS, izmenenie chislennosti pomorskih obshhin. *UWM Olsztyn Acta Neophilologica* XXIV: 277–91. [CrossRef]

*Pravda o russkih raskol'nikah, nazyvajushhihsja staroverami i staroobrjadcami. Po soch. preosvjashh. Makarija, arhim. Pavla (Prusskogo), prof. N.I. Subbotina i dr. Pereskaz D-na N-ja K-va*. 1897. Moskva: tip. I. D. Sytina.

Prugavin, Aleksandr. 1881. Znachenie sektanstva v russkoj narodnoj zhizni. *Russkaja mysl'* 1: 301–63.

Publiskais pārskats par Tieslietu ministrijā iesniegtajiem reliģisko organizāciju pārskatiem par darbību 2005–2021. gadā. 2006–2022, Tieslietu ministrija. Available online: https://www.tm.gov.lv/lv/publikacijas-un-parskati (accessed on 29 May 2023).

*Rasskazy byvshih staroobrjadcev o zhizni v raskole i obrashhenii v Pravoslavie. Vypusk 1*. 1892. Moskva: tip. Je. Lissnera i Ju. Romana.

Robson, Roy R. 1993. Liturgy and Community among Old Believers, 1905–1917. *Slavic Review* 52: 713–24. [CrossRef]

Robson, Roy R. 2004. *Solovki: The Story of Russia Told Through Its Most Remarkable Islands*. New Haven and London: Yale University Press.

Robson, Roy R. 2014. Not Something Ordinary, but a Great Mystery: Old Believer Ritual in the Late Imperial Period. In *Orthodox Christianity in Imperial Russia: A Source Book on Lived Religion*. Edited by H. Coleman. Bloomington: Indiana University Press, pp. 185–91.

Robson, Roy R. 2015. Of Duma or Antichrist: Old Believers and Russian Politics, 1905–1914. In *Church and Society in Modern Russia: Essays in Honor of Gregory L. Freeze*. Edited by M. Hildermeier and E. Wirtschafter. Wiesbaden: Harrassowitz Verlag, pp. 173–84. [CrossRef]

Rogers, Douglas. 2009. *The Old Faith and the Russian Land: A Historical Ethnography of Ethics in the Urals*. Ithaca: Cornell University Press.

Saleniece, Irēna. 2008. *1949. gada 25. martā izvesto balsis*. Daugavpils: Daugavpils Universitātes Akadēmiskais apgāds 'Saule'.

Scheffel, David Z. 1991a. *In the Shadow of Antichrist: The Old Believers of Alberta*. Toronto: University of Toronto Press.

Scheffel, David Z. 1991b. Russian Old Ritualists. *Anthropology Today* 7: 20–21. [CrossRef]

Sementovskij, Aleksandr. 1872. *Jetnograficheskij obzor Vitebskoj gubernii*. Sankt-Peterburg: Tipografija M. Hana.

Shhapov, Afanasij. 1859. *Russkij raskol staroobrjadstva, rassmatrivaemyj v svjazi s vnutrennim sostojaniem Russkoj Cerkvi i grazhdanstvennosti v XVII veke i pervoj polovine VIII. Opyt istoricheskogo issledovanija o prichinah proishozhdenija i rasprostranenija russkogo raskola*. Kazan: Ivan Dubrovin.

Shhapov, Afanasij. 1862. *Zemstvo i raskol. Vyp. 1*. Sankt-Peterburg: Tipografija tovarishhestva «Obshhestvennaja pol'za».

Skorov, Aleksandr. 1903. *Zakony o raskol'nikah i sektantah*. Moskva: Tipo-Litografija I. Pashkova.

Smirnov, Petr. 1898. *Vnutrennie voprosy v raskole v XVII veke*. Sankt-Peterburg: t-vo "Pechatnja S.P. Jakovleva".

Spinka, Matthew. 1941. Patriarch Nikon and the Subjection of the Russian Church to the State. *Church History* 10: 347–66. [CrossRef]

Stanley, Arthur P. 1861. *Lectures on the History of the Eastern Church*. London: J. M. Dent & Co.

*Staroobrjadcheskij cerkovnyj kalendar' na 1949 god*. 1949. Riga: Rizhskaja Grebenshhikovskaja staroobrjadcheskaja obshhina.

*Starovery Litvy: Istorija, kul'tura, iskusstvo*. 2011. Edited by N. Morozova and G. Potashenko. Vil'njus: Mokslo ir enciklopedijų leidybos centras.

Stasulane, Anita. 2021. Identity Multiplicity in an Ethnic and Religious Minority in Latvia: Old Believer Youth. *Frontiers in Sociology* 6: 1–14. [CrossRef]

Subbotin, Nikolaj. 1892. *O sushhnosti i znachenii raskola v Rossii*. Sankt-Peterburg: Sinodal'naja tipografija.

Trofimov, Iosif. 2006. Social'no-psihologicheskij portret starovera v tvorchestve N.S.Leskova 1860-h godov. In *Mezhdunarodnye Zavolokinskie chtenija. Sbornik 1*. Riga: Elpa 2, pp. 195–207.

Ulianova, Galina. 1998. Old Believers and New Entrepreneurs: Religious Belief and Ritual in Merchant Moscow. In *Merchant Moscow: Images of Russia's Vanished Bourgeoisie*. Edited by Edith Clowes, Thomas Owen, James West and Iurii Petrov. Princeton: Princeton University Press, pp. 61–72.

Vysochajshe utverzhdennoe 12 fevralja 1907 g. 1911. Polozhenie Soveta Ministrov ob izdanii vremennyh pravil dlja uzakonenija nezapisannyh v metricheskie knigi brakov staroobrjadcev i sektantov, a takzhe proisshedshego ot sih brakov potomstva. In *Polnyj svod zakonov Rossijskoj imperii. Vse 16 tomov so vsemi otnosjashhimisja k nim Prodolzhenijami i s dopolnitel'nymi uzakonenijami po 1 Nojabrja 1910 goda. V dvuh knigah. Kniga 1*. Edited by A. A. Dobrovol'skij. Sankt-Peterburg: izdanie juridicheskogo knizhnogo magazina I.I. Zubkova pod firmoju «ZAKONOVEDENIE».

Vysochajshe utverzhdennoe 31 janvarja 1907 g. 1911. Polozhenie Soveta Ministrov o predostavlenii posledovateljam staroobrjadcheskih soglasij, ne priznajushhih duhovnyh lic, vozlagat' vedenie svoih knig grazhdanskogo sostojanija na osobyh starost. In *Polnyj svod zakonov Rossijskoj imperii. Vse 16 tomov so vsemi otnosjashhimisja k nim Prodolzhenijami i s dopolnitel'nymi uzakonenijami po 1 Nojabrja 1910 goda. V dvuh knigah. Kniga 1*. Edited by A. A. Dobrovol'skij. Sankt-Peterburg: izdanie juridicheskogo knizhnogo magazina I.I. Zubkova pod firmoju «ZAKONOVEDENIE».

Wallace, Donald Mackenzie. 1961. The Dissenters; Church and State. In *Russia: On the Eve of War and Revolution*. Edited by C. Black. Princeton: Princeton University Press, pp. 407–27, 428–40.

White, James. 2019. 'The Free Sale of Opium': The Reaction of Russian Orthodox Churchmen to Freedom of Conscience, 1864–1905. *European History Quarterly* 49: 203–30. [CrossRef]

Zavarina, Antonina. 1986. *Russkoe naselenie vostochnoj Latvii vo vtoroj polovine XIX—nachale XX veka: Istoriko-jetnograficheskij ocherk*. Zinatne: AN Latv. SSR. Inst. istorii. Riga.

Zavoloko, Ivan. 1929. *Al'bom starinnyh russkih uzorov*. Riga: izdanie akc. obshh. "Salamandra".

Zavoloko, Ivan. 1933. *O staroobrjadcah g. Rigi: Istoricheskij ocherk*. Riga: Rizhskij kruzhok revnitelej stariny pri obshhestve "Grebenshhikovskoe uchilishhe".

Zavoloko, Ivan. 1935. *Svjatye otcy o prazdnovanii Pashi*. Riga: Rizhskaja Grebenshhikovskaja Staroobrjadcheskaja Obshhina.

Zavoloko, Ivan. 1936. *Uchebnik po Zakonu Bozhiju*. Riga: Izdanie Soveta Rizhskoj Grebenshhikovskoj Staroobrjadcheskoj Obshhiny. First published 1933.

Zavoloko, Ivan. 1937. *Duhovnye stihi starinnye. Vyp. 1, vyp. 2*. Riga: Izdanie M. Didkovskago. First published 1933.

Zavoloko, Ivan. 1937. *Istorija Cerkvi Hristovoj*. Riga: Sovet Rizhskoj Grebenshhikovskoj staroobrjadcheskoj obshhiny.

Zavoloko, Ivan. 1939. *Drevnerusskaja vyshivka*. Riga: Izdanie I. Zavoloko.

Zen'kovskij, Sergej. 2006. *Russkoe staroobrjadchestvo. Toma I i II*. Moskva: Institut DI-DIK.

Zhuravljov, Andrej. 1794. *Polnoe istoricheskoe izvestie o staroobrjadcah, ih uchenii, delah i razglasijah*. Sankt-Peterburg: Sinod. tipografija.

