# Peer review of "Church and State and the Marital Rights of Old Believers in Latvia: From Illegality to Secularization"

_religions, doi:10.3390/rel14070839_

Round 1

Reviewer 1 Report

This paper is well written, with a clear and well supported argument and thorough use of sources of various kinds.  The surprising figures on recent marriages in the Latvian Old Believer community given in lines 500-503 (4.4, final paragraph) require a citation, however. Other than that, I recommend the paper for publication without any issues.

The English of this paper is completely intelligible throughout. Minor corrections to grammar, especially regarding the use of articles, are called for frequently, as well as occasional corrections to vocabulary. The paper is easy to read and understand, however.

Author Response

Dear Reviewer,

thank you for your suggestions.

Reviewer 1: The surprising figures on recent marriages in the Latvian Old Believer community given in lines 500-503 (4.4, final paragraph) require a citation, however.

Comments: It was my inattention. I have added the reference in the text; however, it was already included in the reference list.

Reviewer 1: The English of this paper is completely intelligible throughout. Minor corrections to grammar, especially regarding the use of articles, are called for frequently, as well as occasional corrections to vocabulary.

Comments: I have ordered proofreading of my paper in Cambridge Proofreading service, so you will find the corrections of language made by “K. Ploeg, MA” and my own corrections.

The proofreader suggested changing the name of my article to “Church and State and the Marital Rights of Old Believers in Latvia: From Illegality to Secularization” and I think this version is more appropriate. Please, see the proofreader’s comments.

Reviewer 2 Report

The reviewed paper presents a historical overview of the changes both in the Old Believers' approach towards marriage and the state regulations of this issue. As the object of the research is the community of the Old Believers in Latvia, the author has to analyze not only a long period of time, but also the legislation of four subsequent states, which makes the research especially valuable but at the same time is the root of its weaknesses.

First of all, there are serious omissions in the presentation of the Old Believers studies in the 19th century, and the author doesn't mention such important names as Andrei Melnikov, Afanasii Shchapov or Nikolai Leskov. This results in a very simplified and incorrect vision of the publications of that time, as well as a one-sided opinion about intelligentsia's approach to Old Believers as "truly Russian people". My recommendation for the author is to get acquainted especially with Shchapov's and Leskov's writings and re-write the first paragraph of the section 2.

Secondly, the conclusions presented in the last part of the paper (about Old Believers who cannot withstand the challenges of modernity) seem quite superficial. In my opinion, it may be the result of limiting the part of the research which deals with the present times to statistical data (numbers of religious ceremonies) and author's speculations about reasons for their sharp decrease. While interviews with representatives of the Old Believers community would give important first-hand observations, I do not insist on including it into the study. Yet what I recommend is to problematize both the concept of secularization and post-secularism, as well as the idea (expressed in the very first phrase of the paper) about Old Believers being "completely subordinated to the power of the past". It seems that this statement is also simplified and stereotypical.

My other comments are rather minor and refer to specific parts of the paper:

- line 218 - the title "About the Secret of Marriage" - I suppose that "secret" is mistranslated here and it meant to be "sacrament" (тайна - таинство in Russian)

- line 398-399 - if reasons for the divorces "need to be searched in the archival documents", it would be recommended to conduct this search before publishing the article, or to delete this part of the sentence, as it makes an impression of the work in progress.

Last but not least, the paper must be proofread by a native speaker as there are numerous incorrect and awkward syntaxic structures.

Although I am not a native speaker, I noticed several awkward syntaxic structures and punctuation mistakes, which suggests that proofreading made by a native speaker is highly recommended.

Author Response

Dear Reviewer,

thank you for your suggestions. Please, find the response in the attachment.

Round 2

Reviewer 1 Report

The minor revision I requested - the addition of a reference for a single claim - has been fulfilled.  I hope the excellent article sees its way to publication.

Reviewer 2 Report

Thank you very much for corrections and taking my suggestions into account.

I would also like to recommend you adding an information about your field research as the base for conclusions you draw. It will make them sound much more solid and grounded!